# Optimization of Preparation and Preclinical Pharmacokinetics of Celastrol-Encapsulated Silk Fibroin Nanoparticles in the Rat

**DOI:** 10.3390/molecules24183271

**Published:** 2019-09-08

**Authors:** Felicia Onyeabor, Amy Paik, Surya Kovvasu, Baoyue Ding, Jelissa Lin, Md Arif Wahid, Sunil Prabhu, Guru Betageri, Jeffrey Wang

**Affiliations:** 1Department of Pharmaceutical Sciences, College of Pharmacy, Western University of Health Sciences, Pomona, CA 91766, USA; 2Department of Pharmaceutics, College of Medicine, Jiaxing University, Jiaxing 314000, China

**Keywords:** silk fibroin nanoparticles, celastrol, optimized formulation, bioanalysis, pharmacokinetics

## Abstract

Celastrol (CL), a bioactive compound isolated from *Tripterygium wilfordii*, has demonstrated bioactivities against a variety of diseases including cancer and obesity. However, its poor water solubility and rapid in vivo clearance limit its clinical applications. To overcome these limitations, nanotechnology has been employed to improve its pharmacokinetic properties. Nanoparticles made of biological materials offer minimal adverse effects while maintaining the efficacy of encapsulated therapeutics. Silk fibroin (SF) solution was prepared successfully by extraction from the cocoons of silkworms, and a final concentration of 2 mg/mL SF solution was used for the preparation of CL-loaded SF nanoparticles (CL-SFNP) by the desolvation method. A stirring speed of 750 rpm and storage time of 20 h at −20 °C resulted in optimized product yield. A high-performance liquid chromatography (HPLC) method was developed and validated for the analysis of CL in rat plasma in terms of selectivity, linearity, intra-/inter-day precision and accuracy, and recovery. No interference was observed in rat plasma. Linearity in the concentration range of 0.05–5 µg/mL was observed with R^2^ of 0.999. Precision and accuracy values were below the limit of acceptance criteria, i.e., 15% for quality control (QC) samples and 20% for lower limit of quantification (LLOQ) samples. Rats were given intravenous (IV) administration of 1 mg/kg of pure CL in PEG 300 solution or CL-SFNP. The pharmacokinetic profile was improved with CL-SFNP compared to pure CL. Pure CL resulted in a maximum concentration (C_max_) value of 0.17 µg mL^−1^ at 5 min following administration, whereas that for CL-SFNP was 0.87 µg mL^−1^ and the extrapolated initial concentrations (C_0_) were 0.25 and 1.09 µg mL^−1^, respectively, for pure CL and CL-SFNP. A 2.4-fold increase in total area under the curve (AUC_0-inf_) (µg h mL^−1^) was observed with CL-SFNP when compared with pure CL. CL-SFNP demonstrated longer mean residence time (MRT; 0.67 h) than pure CL (0.26 h). In conclusion, the preparation of CL-SFNP was optimized and the formulation demonstrated improved pharmacokinetic properties compared to CL in solution following IV administration.

## 1. Introduction

Pancreatic ductal adenocarcinoma (PDAC) is the major form of malignancy in the pancreas [1]. Unsatisfactory results of conventional chemotherapeutic options demand the search for more effective anticancer agents. Recently, traditional Chinese medicines have received increasing attention in the scientific community for potential clinical applications [2]. 

The Chinese herb *Tripterygium wilfordii* (TW), also known as Lei Gong Teng (literally meaning thunder god vine) has demonstrated many bioactivities. The herbal extract as well as TW, two major bioactive compounds, triptolide (TP) and celastrol (CL, Figure 1), have attracted widespread interest in the research fields of Alzheimer’s disease, rheumatoid arthritis, asthma, hypertension, and systemic lupus erythematosus [3]. Recently, several investigations revealed the potential use of CL in treating obesity [4,5]. Both CL and TP alone and in combination have demonstrated anticancer activity against a number of cancers [6,7]. The anti-proliferative effects of CL are centered on its ability to induce apoptosis in cancer cells derived from the pancreas, lung, glia, prostate, blood, and breast. In addition, CL exhibits the ability to suppress angiogenesis and is a potent proteasome inhibitor [8,9]. However, the clinical use of CL is limited due to its poor solubility and unsatisfactory pharmacokinetic and side effect profiles [8,10]. Recently, biopolymer-based nanoparticles have attracted widespread attention for the use of biocompatible materials, which possess properties of low toxicity and biodegradability. Examples of biopolymer nanoparticles (NPs) include albumin, gelatin, and silk fibroin (SF) [11,12]. SF is a fibrous protein polymer in silk generated from the cocoon of domesticated silkworms, *Bombyx mori*. SF exhibits slow degradation, stealth mechanical properties, favorable processability, good biodegradability, and regenerative properties, which makes this biomaterial suitable in biomedicine and nanomedicine [12,13,14,15,16]. The protein structure of SF is comprised of a heavy chain (~325 kDa) and a light chain (~125 kDa). The secondary structure of SF consists of re-oriented β-sheets, which make up the hydrophobic block of SF. Hydrogen bonds in the β-sheet domain contribute to the strength and stability of the fiber in SF. The hydrophilic block is the amorphous domain of the SF structure, which contributes to the strength and stability of the fiber [12,13,14,15,16].

We previously prepared CL-loaded SF nanoparticles (CL-SFNP) using a desolvation method [17]. In vitro drug release studies of CL-SFNP revealed a slow and sustained release of drug at physiological pH (pH 7.4) and rapid release at lysosomal pH (pH 4.5). The in vitro cytotoxicity study against two human pancreatic cancer cell lines (PANC-1 and Mia PaCa-2) demonstrated increased growth inhibition with the nanoparticle formulation compared to pure CL alone [17]. 

The main objectives of this study were 1) to optimize the preparation of CL-SFNP; 2) to develop and validate a HPLC method for the analysis of CL in rat plasma; and 3) to determine the pharmacokinetic profile of optimized formulation of CL-SFNP in comparison to pure CL in male Sprague–Dawley rats. 

## 2. Results 

### 2.1. Preparation of Silk Fibroin Solution 

Concentration of the SF solution was measured by freeze-drying 5 mL of the solution (in triplicate) followed by weighing the material. The concentration of the solution was 4.64 ± 0.02 mg/mL, which was further diluted with deionized water to 2 mg/mL to give the optimized concentration for nanoparticle formation as previously reported [17]. Consistency in SF solution preparation was optimized by a stirring time of 1 h during the hydrolysis step of the SF material. 

### 2.2. Optimized Preparation of CL-Loaded Silk Fibroin Nanoparticles 

#### 2.2.1. Varying Storage Times at −20 °C

Storage time affects the formation and stability of SFNP [18,19]. For this study, a wider range of storage times were evaluated. The percent yield of the lyophilized CL-SFNP product was the main indicator of quantifying the effect of different storage times at −20 °C. The size and zeta-potential were measured to characterize the nanoparticles. Table 1 shows the percent yield, size, and zeta-potential that corresponds to eight different storage times (1, 2, 4, 8, 16, 20, 24, and 48 h). Low percent yields were observed for storage times of 1, 2, and 48 h (1.52 ± 0.5%, 3.89 ± 1.3%, and 4.54 ± 2.1%, respectively), with significantly larger size for 1 and 48 h (1390.2 ± 84 nm and 4809.9 ± 7132.2 nm, respectively). Higher percent yields were observed for storage times from 4 to 20 h with the highest yield of 35.5% ± 7.5% at 20 h. 

#### 2.2.2. Varying Rotation Speeds 

Stirring speed has an impact on β-sheet formation, which is responsible for the stability and mechanical properties of the SF material and in turn may affect the formation and stability of nanoparticles [13,16]. The stirring speed during the addition of 2 mg/mL CL into 2 mg/mL SF was observed at a slightly lower speed of 625 rpm and a higher speed of 875 rpm with 20 h as the selected storage time at −20 °C. The percent yield for stirring speed 625 rpm was 27.1% ± 4.7%, which is slightly lower than that of the set speed of 750 rpm. Increasing the speed to 875 rpm resulted in a much lower percent yield of 18.4% ± 7.2% compared to that for the 750 rpm stirring speed (Table 1). 

#### 2.2.3. Inter-Day Evaluation of Optimized Formulation 

Storage time of 20 h at −20 °C and a stirring speed of 750 rpm were selected based on the product yield. Inter-day evaluation was performed on three consecutive days with full characterization of the nanoparticles as given (Table 2). The product yield was 26.5% ± 8.7%, 31.1% ± 16.9%, and 34.3% ± 14.5% for days 1, 2, and 3, respectively. Therefore, CL-SFNP could be prepared consistently using the optimized procedures. The general flow chart of CL-SFNP preparation is shown in Figure 2.

### 2.3. Analytical Method Validation 

#### 2.3.1. Chromatographic Conditions 

Pristimerin (Figure 1) was selected as the internal standard (IS) based on its similarity in structure and ultraviolet (UV) absorption spectrum to CL. The maximum UV absorbance of CL and pristimerin was at 425 nm [20]. The mobile phase was a mixture of acetonitrile and 1% phosphoric acid (H_3_PO_4_) (85:15 v/v). Acidification of the mobile phase was to prevent CL from ionizing and improve the peak shape. Retention times of CL and pristimerin were 4.9 and 9.4 min, respectively.

#### 2.3.2. Selectivity and Specificity 

The potential inferences of biological matrix with both the analyte and IS were assessed in blank rat plasma, rat plasma spiked with IS, CL, or SFNP. All samples were processed for HPLC analysis with no interference observed (Figure 3). 

#### 2.3.3. Linearity and Sensitivity 

The calibration curve demonstrated linearity in the concentration range of 0.05–5 µg/mL, with an R^2^ value greater than 0.99. Sensitivity was defined as the lowest concentration that could be quantified in accordance to precision and accuracy [21]. The lowest level of quantification (LLOQ) was found at 0.05 µg/mL. 

#### 2.3.4. Precision and Accuracy 

The intra- and inter-day precision and accuracy were evaluated at the lower limit of quantification (LLOQ), low quality control (LQC), medium quality control (MQC), and high quality control (HQC) concentrations as indicated in Table 3. The intra-/inter-day precision of the quality control (QC) samples defined by relative standard deviation (RSD) ranged from 2.60% to 14.33% and 1.94% to 17.60% for LLOQ samples, which was within the limit of the acceptance criteria of %RSD ≤ 15% for QC samples and ≤20% for LLOQ. The accuracy ranged from 90.22% to 114.26% for QC samples and 110.75% to 119.28% for LLOQ, where the values did not deviate from the nominal concentration by ±15% for QC samples and ±20% for LLOQ [21].

#### 2.3.5. Recovery 

The method used for extraction recovery was protein precipitation with acetonitrile. Table 4 shows the percent recovery of CL and IS. The recovery of CL was above 40% ranging from 40.95% to 67.43%, and that of the IS was 75.28%. Although the recovery of CL was relatively low, it was reproducible, which meets FDA guideline [21].

### 2.4. Pharmacokinetic Study

From previous studies, CL has demonstrated poor pharmacokinetic profile with low bioavailability and rapid clearance [3]. Nanocarriers have been used to deliver problematic pharmaceuticals, by improving drug absorption, distribution, metabolism, and excretion [22,23]. In this study, the optimized CL-SFNP was evaluated compared to pure CL in solution following IV administration. The plasma concentration–time curves shown in Figure 4 demonstrated an improvement in pharmacokinetic behavior upon SF nanocapsulation. The CL initial concentration (C_0_) following CL-SFNP administration was four times higher than that of CL in solution in rats. Similarly, the total area under the curve (AUC_0-inf_) value with the nanoformulation was 2.5 times higher and had a mean residence time (MRT) that was twice as long compared to those with CL in solution. The relevant pharmacokinetic parameters are shown in Table 5. The circulation time of CL in blood was longer with the NP formulation when compared to that with CL in solution. Our findings align with previous reports on the improvement of pharmacokinetic profiles with various CL-NPs [20,24,25,26]. 

## 3. Discussion

The pharmacokinetic properties of CL have rarely been reported mainly because of its low concentration in plasma which is due to low solubility (Song et al., 2011) and extensive in vivo metabolism [3]. CL is undissolvable in water, with a LogP of 5.63 which qualifies CL as a Biopharmaceutics Classification System (BCS) Class IV-type drug, suggesting CL may be poorly absorbed by the body [27]. To overcome these limitations, nanotechnology has been employed to improve both physiochemical and pharmacokinetic properties of potential compounds. The protein material employed for this study was SF, which is known for its exceptional mechanical durability and biocompatibility. The fabrication of CL-SFNP was generated by the desolvation method, where the drug dissolved in organic solvent was introduced, dropwise, into SF solution under stirring. 

Before the application of CL-loaded SF nanoparticles for in vivo evaluation, adjustments were made to optimize the product in regard to varying storage time and rotation speed, followed by inter-day evaluation of the finalized formulation. A stable SF solution that yielded greater NP product was observed when the SF material was stirred for 1 h as opposed to 30 min. Of the range of storage times, 20 h was selected as it presented the most percent yield of CL-SFNP product of 35.5% ± 7.5%, despite the slightly larger size than at 4 h. CL-SFNP was negatively charged. A stirring speed of 750 rpm was found favorable and was chosen to be suitable for further studies. The characteristics of the nanoparticles were consistent from three batches prepared on three consecutive days. The optimized procedure for CL-SFNP preparation was used to generate product that was used in the pharmacokinetic study.

A high-performance liquid chromatography method was developed and validated for the analysis of CL in rat plasma in terms of selectivity, linearity, intra-/inter-day precision and accuracy, and recovery. The HPLC method was validated using Food and Drug Administration (FDA) guidelines with regard to linearity where the R^2^ value was greater than 0.99, intra-/inter-day precision and accuracy values with a percent deviation below 15% for QC samples and 20% for LLOQ, and a consistent recovery. The validated method was applied for the analysis of CL in plasma to evaluate the pharmacokinetic behavior of CL-SFNP in comparison with CL solution. 

The pharmacokinetic profile was improved with CL-loaded SF nanoparticles compared to pure CL—which was observed following IV administration. The initial concentration of CL nanoparticles was four times higher than that of CL in solution and CL-SFNP demonstrated longer mean residence time. Future research will focus on the evaluation of bioactivities of CL-SFNP in comparison with CL, with a long-term aim of developing CL-SFNP for treating diseases such as PDAC, autoimmune diseases, and obesity.

## 4. Materials and Methods 

### 4.1. Chemicals and Reagents 

Celastrol (CL) was purchased from Chengdu Biopurify Phytochemicals Ltd. (Chengdu, China). Cocoons from *B. mori* silkworms were kindly supplied by Tongxiang Mulberry Silk Base of Zhejiang Province (Tongxiang, China). A dialysis membrane with a cut-off of 7000 Da molecular weight cut-off (MWCO) was obtained from Viskase Companies, Inc. (Chicago, USA). Pristimerin purchased from PI & PI TECH (Macon, GA, USA). HPLC-grade acetonitrile and methanol used for the analysis were purchased from VWR (Randor, PA, USA). HPLC-grade phosphoric acid was obtained from Fisher Scientific (Pittsburgh, PA, USA). Ultrapure water was obtained from Millipore equipment (Millipore, Bedford, MA, USA). Rat plasma was purchased from Innovative Research (Novi, MI, USA). Polyethylene glycol 300 was purchased from Spectrum (Gardena, CA, USA). Dulbecco’s phosphate buffered saline was purchased from Genesse Scientific (San Diego, CA, USA). 

### 4.2. Preparation of Silk Fibroin Solution 

One gram of cut *B. mori* cocoon was degummed twice in 0.02 M sodium carbonate (Na_2_CO_3_) for 30 min with rinses with deionized water between degumming steps. The extracted SF was allowed to dry overnight at room temperature. Dried SF was then hydrolyzed in 50 mL of a ternary system containing calcium chloride dihydrate (CaCl_2_·2H_2_O):ethanol:water (molar ratio of 1:2:6) under constant stirring for 1 h until SF was completely dissolved, followed by an additional 3 h without stirring with temperature maintained at 75 °C. The hydrolyzed SF was transferred into 50 mL centrifuge tubes and centrifuged for 15 min at 4500 rpm to remove any undissolved material and the supernatant was subsequently dialyzed against deionized water under gentle stirring (250 rpm) for the next 72 h. Water was changed after 1 and 3 h on day 1 and twice a day for day 2 and day 3. The dialyzed SF solution was collected and centrifuged for 15 min at 14,000 rpm to remove any nanoparticles that might have formed during the dialysis. The supernatant was collected and a 5 mL sample was placed into pre-weighed vials, and lyophilized to determine the concentration of the SF solution [17,18]. Based on the result, the SF solution was diluted to 2 mg/mL. 

### 4.3. Preparation of CL-Loaded Silk Fibroin Nanoparticles 

For the preparation of CL-SFNP, 0.4 mL of 2 mg/mL of CL (CL) dissolved in acetone:ethanol (3:2 v/v) was added dropwise into 4 mL of 2 mg/mL SF solution under gentle stirring at different stirring speeds of 625, 750, and 875 rpm. The addition of CL into SF solution was completed in the span of 2 min and the mixture was allowed to stir for 10 min, followed by sonication for 3 min. The suspension was then stored at −20 °C for a fixed time and quickly thawed at 35 °C. The thawed suspension was then subjected to ultracentrifugation at 14,200 rpm for 30 min. The pellet was separated and washed twice with deionized water and the nanosuspension was further dispersed by ultrasound processor set at 10% amplitude for 1 min. The suspension was then frozen at −80 °C followed by lyophilization. The lyophilized product was weighed and percent recovery was determined [17,28,29]. All the steps are shown in Figure 2.

### 4.4. Nanoparticle Characterization

#### 4.4.1. Size and Zeta-potential 

To one milligram of freeze-dried CL-SFNP was added 2 mL of deionized water to obtain a concentration of 0.5 mg/mL of CL-SFNP. The suspension was then dispersed by an ultrasound processor set at 10% amplitude for 5 min. The dispersed suspension was further diluted (1:5) with deionized water, followed by analysis with NanoBrook^®^ (Brookhaven Instruments Corporation, NY, USA) to measure the size of CL-SFNP. The surface charge of CL-SFNP was measured by Zetasizer^®^ (Nano ZS90, Malvern, Malvern Instruments Ltd., Worcestershire, UK).

#### 4.4.2. Encapsulation Efficiency and Drug Loading 

To measure the drug content of CL in SF nanoparticles, 2 mL of acetone:ethanol (3:2 v/v) was added to 0.5 mg of freeze-dried product to dissolve the drug. The mixture was centrifuged at 14,000 rpm for 15 min and the supernatant was collected and used for HPLC analysis. As for the encapsulation efficiency, the supernatant and washes that were collected during the CL-SFNP preparation, was centrifuged at 14,000 rpm for 15 min and the supernatant was used for HPLC analysis. 

Encapsulation efficiency and drug loading was measured by Agilent 1260 infinity HPLC systems (Agilent Technologies, Palo Alto, CA, USA). Chromatographic separation was carried out on an Accucore XL C_18_ column (250 mm × 4.6 mm, 4 µm). Analysis was performed with UV detection, using a diode array detector (DAD) set at 425 nm for CL. The mobile phase consisted of 1% (v/v) phosphoric acid (A) and methanol (B), programmed for a gradient elution, where the mobile phase (B) increased from 58% to 95% during 4 min and remained at 95% for 8 min. The gradient elution ran at a flow rate of 1 mL/min and the acquisition time was 12 min. Encapsulation efficiency and drug loading were calculated using the following equations.
(1)$$Encapsulation\;efficiency\;=\frac{Total\;weight\;of\;feeding\;drug\;\left(mg\right)-Amount\;of\;drug\;analyzed\;by\;HPLC\;method\;\left(mg\right)}{Total\;weight\;of\;feeding\;drug\;\left(mg\right)}x100\%\;$$
(2)$$Drug\;loading\;equation\;=\;\frac{\;Amount\;of\;drug\;analyzed\;by\;HPLC\;method\;\left(\mu g\right)}{Weight\;of\;nanoparticles\;\left(mg\right)}\;$$

### 4.5. Optimized Formulation of CL-Loaded Silk Fibroin Nanoparticles 

#### 4.5.1. Varying Storage Times at −20 °C

To investigate a suitable storage time for CL-SFNP at −20 °C, the preparations were stored at −20 °C for 1, 2, 4, 8, 16, 20, 24, and 48 h. After storage for the indicated times the CL-SFNP were further purified following the procedures mentioned above. 

#### 4.5.2. Varying Rotation Speeds

The rotation speed at time of loading CL into SF solution was set at 625, 750, or 875 rpm. The drug-loaded nanoparticles were stored for the optimized time of 20 h at −20 °C. 

#### 4.5.3. Inter-Day/Intra-Day Evaluation of the Finalized Formulation 

To confirm the integrity of the finalized formulation of CL-SFNP, the product yield was observed for three consecutive days and was done in triplicate. The lyophilized product was weighed and further characterized in terms of size, zeta-potential, encapsulation efficiency, and drug loading. 

### 4.6. Development and Validation of Method for Preclinical Studies

To ensure that the HPLC method developed for the analysis of CL in rat plasma for pharmacokinetic studies was reliable and reproducible, the method was validated in terms of specificity and selectivity, linearity, precision and accuracy, recovery, and stability [21]. 

#### 4.6.1. Chromatographic Conditions 

Chromatographic separation was carried out on an Accucore XL C_18_ column (250 mm × 4.6 mm, 4 µm) using an Agilent 1260 Infinity HPLC system. Analysis was performed with ultraviolet (UV) detection using a diode array detector (DAD) set at 425 nm for CL and pristimerin (IS). The mobile phase consisted of acetonitrile and 1% H_3_PO_4_ (85:15 v/v), with a flow rate set at 1 mL/min and column temperature set at 25 °C. The acquisition time was 15 min. 

#### 4.6.2. Preparation of Standard and Quality Controls

CL and internal standard stock solutions were prepared at concentration of 1 mg/mL in methanol. The CL calibration standard concentrations of 0.05, 0.1, 0.2, 0.5, 1, and 5 µg/mL were prepared in blank rat plasma. Quality control (QC) samples were set at low (0.1 µg/mL), medium (0.5 µg/mL), and high (5 µg/mL) concentrations, and were prepared the same as standard concentrations. All standard and quality control samples were spiked with 10 µL IS of concentration 20 µg/mL.

#### 4.6.3. Preparation of Plasma Samples 

Plasma samples (100 µL each) were spiked with 10 µL of IS. The spiked plasma samples were extracted using the protein precipitation method with 400 µL of acetonitrile. The precipitated sample was vortexed for 1 min and then underwent centrifugation for 5 min at 14,000 rpm. The supernatant was transferred into glass tubes and the solvent was removed by evaporation with a gentle stream of air for 2.5 h. The dried residue was reconstituted with 50 µL of the mobile phase by vortexing for 1 min and centrifuging for 5 min at 14,000 rpm. Supernatant volume of 35 µL was collected and transferred into vials for HPLC analysis.

#### 4.6.4. Specificity and Selectivity 

Absence of interference with the matrix and the analyte was ascertained for six sources. Selectivity was evaluated by injecting the blank plasma, plasma spiked with CL or IS, plasma spiked with both CL and IS, or plasma spiked with blank SFNP formulation. 

#### 4.6.5. Linearity and Sensitivity 

Linearity was evaluated by injecting a standard curve with six concentrations of CL ranging from 0.05 to 5 µg/mL. A calibration curve was constructed using CL peak area to IS peak area (ratio CL/IS) and plotting it versus the nominal concentrations of CL. Linearity of the curve was generated by using linear least square regression (R^2^) analysis. Sensitivity was defined by the lowest concentration of the curve that can be measured and quantified in terms of precision and accuracy. The CL concentration of 0.05 µg/mL was determined as the LLOQ.

#### 4.6.6. Precision and Accuracy 

Standard and quality control samples were evaluated in terms of intra-day precision and accuracy in replicates of five on a single day and for inter-day precision and accuracy over three consecutive days. Relative standard deviation (RSD%) was determined for precision and was acceptable of all quality control (QC) samples which did not exceed 15% and 20% for LLOQ samples under the acceptance criteria set by FDA guidelines. Accuracy was determined by comparing the calculated concentrations derived from the equation form the calibration curve to the nominal concertation. The acceptance criteria for accuracy was the mean values of calculated concentration compared to nominal concentration fell within ±15% for QC samples and ±20% for LLOQ.

#### 4.6.7. Recovery

The extraction recovery of CL and IS in rat plasma was compared to non-extracted standard solution containing same amount of both analyte and internal standard. The recovery was evaluated at all levels of QC samples in replicates of five (0.1, 0.5, and 5 µg/mL) for CL and for IS concentration of 2 µg/mL. The process to determine extraction has been mentioned above and compared with non-extracted samples represented as analyte and IS in solution. 

### 4.7. Pharmacokinetics 

The study on animal protocol was approved by Western University of Health Sciences, Pomona Institutional Animal Care and Use Committee (IACUC). Male Sprague–Dawley rats with pre-cannulated jugular vein were administered an intravenous (IV) dose of 1 mg/kg. Rats were separated into two groups (n = 3) and were administered pure CL solution in PEG 300 or CL-SFNP suspension in phosphate buffer solution (PBS) (Figure 4) into the pre-cannulated jugular vein. After IV administration, 200 µL of blood was collected into pre-heparinized tubes at 0.083, 0.5, 1, 2, 4, 6, 8, 12, and 24 h. After 24 h the rats were euthanized with 30% isoflurane. Collected blood was then centrifuged for 15 min at 14,000 rpm. After centrifugation 110 µL of plasma was collected and transferred into Eppendorf tubes, and then stored at −20 °C until analysis. The pharmacokinetic parameters were generated by non-compartmental analysis using PK Functions for Microsoft Excel [30].

## 5. Conclusions 

The optimization of CL-SFNP with respects to storage time at −20 °C and rotation speeds was able to be evaluated where a greater product yield of CL-SFNP was collected and the selected parameters were subjected to pharmacokinetic analysis. The HPLC method was developed and validated and further used in the analysis of plasma concentration of CL for pharmacokinetic studies. The CL-SFNP demonstrated an improvement of the pharmacokinetic profile of CL compared to pure CL. 

## Figures and Tables

**Figure 1 molecules-24-03271-f001:**
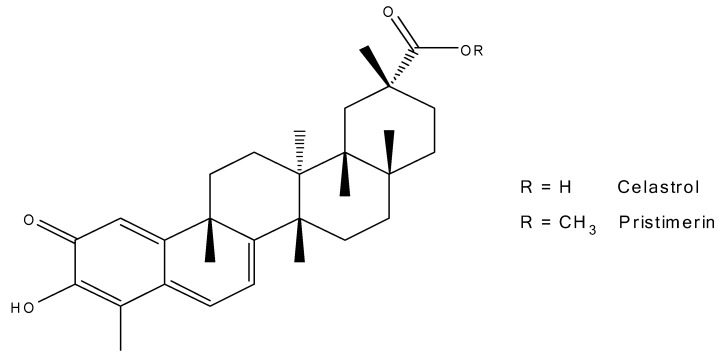
Chemical structures of celastrol (CL) and pristimerin.

**Figure 2 molecules-24-03271-f002:**
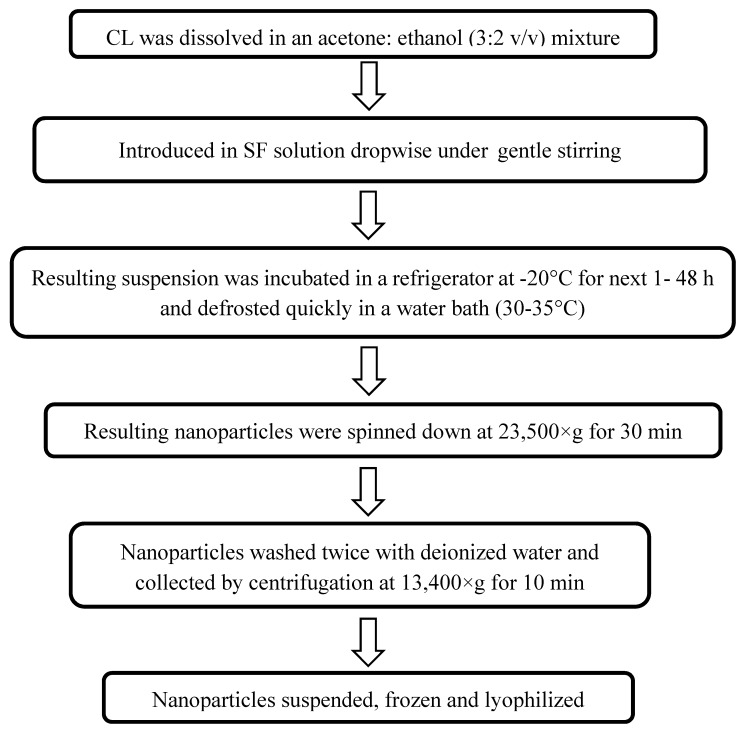
Flow diagram of preparation of CL encapsulated silk fibroin (SF) nanoparticles.

**Figure 3 molecules-24-03271-f003:**
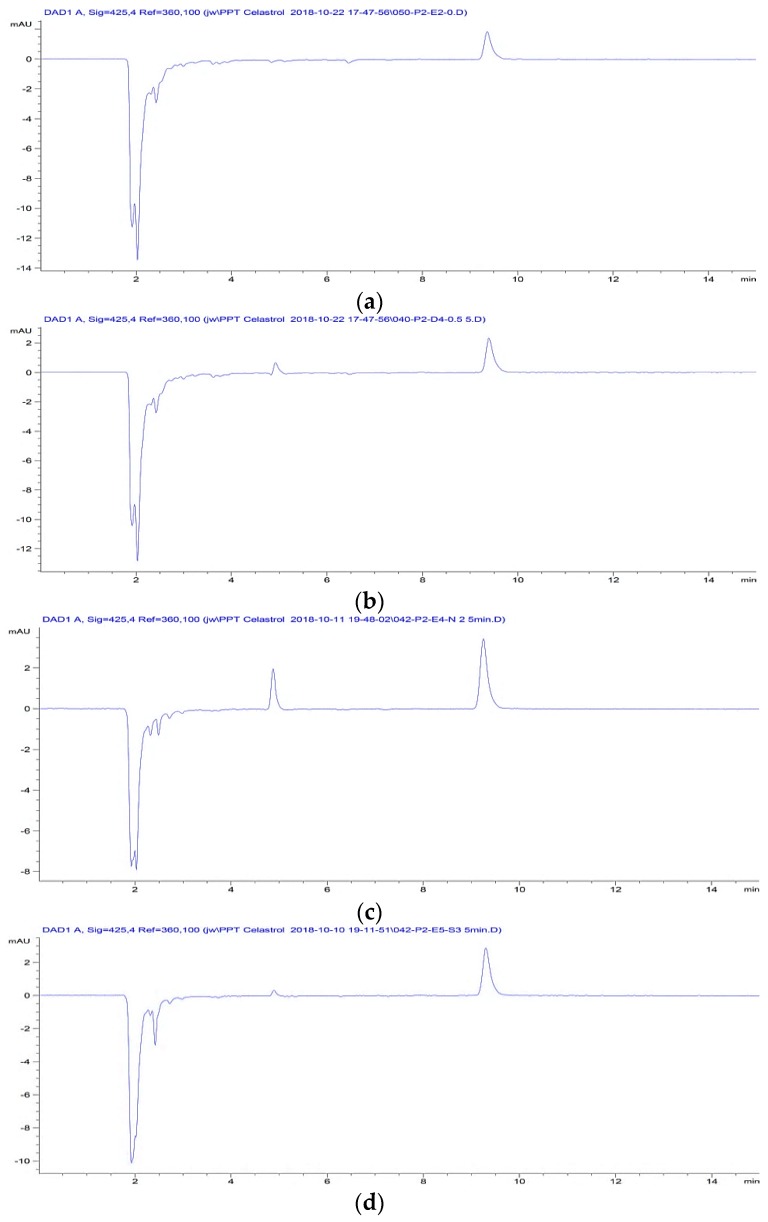
Chromatograms of processed samples of (**a**) Blank plasma spiked IS (2 µg/mL) (**b**) Rat plasma spiked with CL (0.5 µg/mL) and IS (2 µg/mL) (**c**) Rat plasma at 5 min after IV dosing of CL-SFNP (1 mg/kg). (**d**) Rat plasma at 5 min after IV dosing of CL-solution (1 mg/kg).

**Figure 4 molecules-24-03271-f004:**
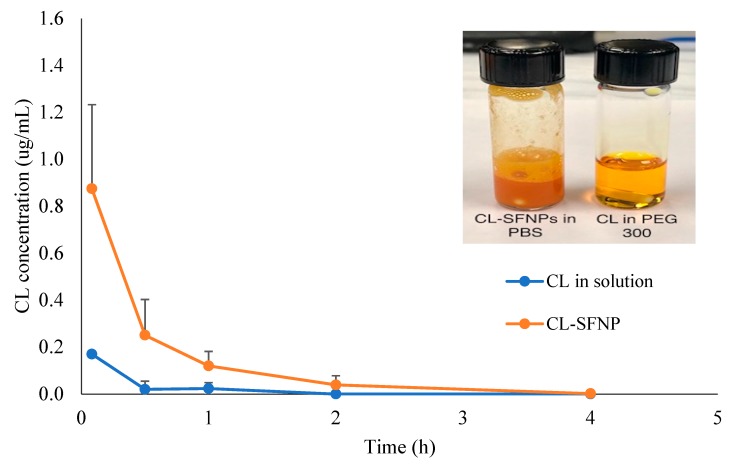
CL plasma concentration–time profiles in rats following intravenous (IV) administration of CL and celastrol-loaded silk fibroin nanoparticles (CL-SFNP). The insert shows the physical appearance of the respective formulations.

**Table 1 molecules-24-03271-t001:** Effect of varying storage times and rotation speeds on percentage yield and size (n = 3).

Hours in −20 °C	Rotation Speed (rpm)	Yield (%) (n = 3)	Size (nm) (n = 3)
1	750	1.52 ± 0.5	1390.2 ± 847.6
2	750	3.89 ± 1.3	424.1 ± 32.4
4	750	33.3 ± 6.9	266.1 ± 2.3
8	750	26.9 ± 8.1	321.2 ± 5.5
16	750	31.9 ± 3.1	277.1 ± 1.5
20	750	35.5 ± 7.5	333.5 ± 16.5
24	750	25.2 ± 2.9	312.1 ± 1.4
48	750	4.54 ± 2.1	4809.9 ± 7132.2
20	625	27.1 ± 4.7	256.2 ± 13.1
20	750	35.5 ± 7.5	333.5 ± 38.8
20	875	18.4 ± 7.2	292.7 ± 28.1

**Table 2 molecules-24-03271-t002:** Inter-day evaluation for physical properties of optimized formulation (n = 3).

Day	Yield (%)	Size (nm)	Zeta-Potential (mV)	Encapsulation Efficiency (%)	Drug Loading (µg/mg)
1	26.5 ± 8.7	291.1 ± 22.1	−25.2 ± 3.2	81.4 ± 8.2	97.2 ± 28.4
2	31.7 ± 16.9	298.9 ± 20.7	−19.5 ± 5.4	83.2 ± 14.7	80.9 ± 20.0
3	34.3 ± 14.5	299.5 ± 20.3	−22.3 ± 0.9	79.3 ± 11.3	80.9 ± 4.2

**Table 3 molecules-24-03271-t003:** Intra-day, inter-day precision, and accuracy of nominal concentrations.

Nominal Concentration (µg/mL)	Intra-Day (n = 5)		Inter-Day (n = 3)	
	Precision (% RSD)	Accuracy (%)	Precision (% RSD)	Accuracy (%)
0.05 (LLOQ)	17.6	110.7	10.8	116.3
0.1	4.6	114.2	7.5	107.8
0.5	4.3	92.2	10.8	97.2
5.0	5.3	101.5	4.7	95.8

RSD—relative standard deviation; LLOQ—lowest level of quantification.

**Table 4 molecules-24-03271-t004:** Recovery of CL and internal standard (IS) from rat plasma (n = 5).

Analyte	Concentration (µg/mL)	Recovery (%)
CL	0.1	67.4 ± 5.3
	0.5	42.0 ± 2.7
	5.0	41.0 ± 5.5
Pristimerin (IS)	2.0	75.3 ± 8.8

**Table 5 molecules-24-03271-t005:** Pharmacokinetic parameters of CL following single intravenous dose of pure CL and CL-SFNP at 1 mg/kg (n = 3).

Parameters	CL in PEG 300	CL-SFNP
C_0_ (µg mL^−1^)	0.25	1.09
AUC_0–inf_ (µg h mL^−1^)	0.18 ± 0.31	0.47 ± 0.20
CL (mL h^−1^)	1.78 ± 0.94	0.71 ± 0.33
MRT (h)	0.26 ± 0.44	0.51 ± 0.07

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
