# Peer review of "Optimization of Preparation and Preclinical Pharmacokinetics of Celastrol-Encapsulated Silk Fibroin Nanoparticles in the Rat"

_molecules, 2019, doi:10.3390/molecules24183271_

Round 1

Reviewer 1 Report

Onyeabor et al. presented the results of their study, focused on the optimization of celastrol-SF nanoparticles, and preclinical pharmacokinetics of these particles in a rat model.

The aim of the study is presented. The introduction includes all the relevant information on the active entity, as well as on silk fibroin. It allows the reader to understand the scientific background of the research. 

The authors should address several minor issues:

1) is the particle size sufficient to exert the pharmacological effect? Some authors suggest that the particle size for SF nanoparticles is 100 nm or below.

2) What was the blood volume collected from rats at each time point?

3) Please, provide the name and manufacturer of the software used for PK calculations. Also, please provide details on the calculation of PK parameters. For example - how was the elimination constant calculated? As the authors present in Figure 4 - the concentrations of CL are extremely low after IV administration of CL in PEG. It makes pharmacokinetic calculations challenging and possibly erroneous.

4) Figure 3 is probably captioned incorrectly. Please revise and correct if necessary.

5)  The conclusions unnecessarily include possible future directions. The authors could state it in the discussion section, rather than in the conclusions. 

Author Response

Reviewer 1:

The aim of the study is presented. The introduction includes all the relevant information on the active entity, as well as on silk fibroin. It allows the reader to understand the scientific background of the research.

The authors should address several minor issues:

1) Is the particle size sufficient to exert the pharmacological effect? Some authors suggest that the particle size for SF nanoparticles is 100 nm or below.

Response: We do agree with the Reviewer 1 comment that a more strict definition of nanoparticles is for those with a particle size less than 100 nm. But there are many reports in the literature using nanoparticles with larger sizes. We recently published a review article on recent advances in nanoformulation-based therapies for pancreatic cancer and many cited studies were conducted with particles larger than 100 nm (Crit Rev Ther Drug Carrier Syst. 2019;36(1):59-91). In our previously published study, these celastrol encapsulated silk fibroin nanoparticles did show enhanced pharmacological effect in comparison with celastrol itself (Nanoscale. 2017 Aug 17;9(32):11739-11753).

2) What was the blood volume collected from rats at each time point?

Response: The blood volume was 200 µL. This information has been added to the manuscript.

3) Please, provide the name and manufacturer of the software used for PK calculations. Also, please provide details on the calculation of PK parameters. For example - how was the elimination constant calculated? As the authors present in Figure 4 - the concentrations of CL are extremely low after IV administration of CL in PEG. It makes pharmacokinetic calculations challenging and possibly erroneous.

Response: PK Functions for Microsoft Excel was used for pharmacokinetic analysis. The program was designed by Joel I. Usansky, Ph.D., Atul Desai, M.S. and Diane Tang-Liu, Ph.D., Department of Pharmacokinetics and Drug Metabolism, Allergan, Irvine, CA 92606, U.S.A. A reference of using this program has been added.

We agree with the reviewer’s comment that the concentrations of celastrol were extremely low after IV administration of celastrol in PEG and it makes pharmacokinetic calculations challenging and possibly erroneous. Therefore, non-compartmental instead of compartmental analysis was conducted. 

4) Figure 3 is probably captioned incorrectly. Please revise and correct if necessary.

Response: Thanks for identifying the error. We have revised the figure order.

5) The conclusions unnecessarily include possible future directions. The authors could state it in the discussion section, rather than in the conclusions.

Response: Thanks for the suggestion. The future direction party has been moved to the discussion section.

Reviewer 2 Report

Considering your paper reported about preclinical pharmacokinetics of celastrol encapsulated silk fibroin nanoparticles, my minor comment on this study is that the manuscript does not provide a convincing case regarding the really new contributions of pharmacokinetic study.

Author Response

1) Considering your paper reported about preclinical pharmacokinetics of celastrol encapsulated silk fibroin nanoparticles, my minor comment on this study is that the manuscript does not provide a convincing case regarding the really new contributions of pharmacokinetic study.

Response: Thanks for your comments. In this manuscript we report three major areas of investigations: Part 1: CL-loaded SF nanoparticles preparation and formulation optimization including stirring time and storage periods’ Part II: Development and validation of a high-performance liquid chromatography method for the analysis of CL in rat plasma in terms of selectivity, linearity, intra/inter-day precision and accuracy, and recovery; and Part 3: Pharmacokinetic  comparison of celastrol in PEG 300 solution and celastrol nanoparticles following intravenous admonition in rats.

The pharmacokinetic profile was improved with celastrol loaded SF nanoparticles compared to pure CL. We strongly believe this information is of a great use in future studies, especially in determining dose and dosing regimens in efficacy and toxicity evaluations.